# Undifferentiated Pancreatic Carcinomas, Clinical Features and Therapeutic Options: What We Know

**DOI:** 10.3390/cancers14246102

**Published:** 2022-12-12

**Authors:** Joseph Mugaanyi, Changjiang Lu, Jing Huang, Caide Lu

**Affiliations:** 1School of Medicine, Ningbo University, Ningbo 315211, China; 2Department of Hepato-Pancreato-Billiary Surgery, Ningbo Medical Center Li Huili Hospital, Ningbo University, Ningbo 315211, China

**Keywords:** undifferentiated carcinoma, pancreatic cancer, treatment, giant cell tumor, prognosis

## Abstract

**Simple Summary:**

Undifferentiated pancreatic carcinomas are a very rare form of malignant pancreatic cancer that are difficult to diagnose and very aggressive. They are a kind of anaplastic pancreatic cancer that accounts for less than 10% of all pancreatic cancers. The WHO classifies undifferentiated pancreatic cancers into osteoclast-like giant cell type, sarcomatoid type and rhabdoid type. Patients may have KRAS, TP53 and SMAD4 alterations. In some cases, INI1 expression may be lost. Although there are no standard guidelines for treatment, in patients with resectable tumors, adjuvant therapy may be also beneficial. Since the prognosis is very poor in patients with unresectable tumors, careful assessment of resectability is very important.

**Abstract:**

Undifferentiated pancreatic carcinomas are rare malignant tumors of the pancreas that are very aggressive and challenging to diagnose. The WHO categorizes them into undifferentiated osteoclast-like giant cell, sarcomatoid, and rhabdoid pancreatic carcinomas. Patients present with nonspecific symptoms such as jaundice, vague abdominal or back pain and itchy skin. Their histological characteristics include positive pan-cytokeratin mononuclear pleomorphic cells, osteoclast-like giant cells and CD68. Patients may have KRAS, TP53 and SMAD4 alterations, homozygous deletions of CDKN2A and CDKN2B, as well as INI1 deficiency. Surgical resection is the only curative treatment. Patients may benefit from postoperative adjuvant therapy. There are no widely accepted guidelines specific to this type of tumor; however, some chemotherapy regimens may be promising. The patient prognosis is mostly poor, especially in patients with unresectable tumors. However, several studies have shown patients achieving long-term survival with adjuvant therapy. In conclusion, although undifferentiated pancreatic carcinoma is rare and very aggressive, there is still potential for improved patient survival with proper diagnosis and treatment.

## 1. Introduction

Pancreatic cancers are known to be very aggressive and have a high mortality rate and an increasing prevalence [1]. Undifferentiated pancreatic carcinoma is a catch-all term for a subset of pancreatic carcinomas characterized by a lack of glandular differentiation and prominent histiocyte and osteoclast-like giant cell infiltration [2,3]. The World Health Organization (WHO) categorizes undifferentiated pancreatic carcinomas under ductal adenocarcinoma, a subset of malignant epithelial tumors of the pancreas [4]. However, they are histologically distinct from pancreatic ductal adenocarcinomas (PDAC). The WHO further divided undifferentiated pancreatic carcinomas into undifferentiated osteoclast-like giant cell, sarcomatoid, and rhabdoid pancreatic carcinomas [4]. These tumors have three histological classifications: pleomorphic, osteoclastic and mixed-type [5,6]. In the literature, they are mostly reported as undifferentiated carcinomas with osteoclast-like giant cells (UCOGC) [3,7,8,9,10,11,12,13,14,15,16,17,18,19], giant cell pancreatic carcinoma [20], sarcomatoid carcinoma of the pancreas [21,22], or rhabdoid pancreatic carcinomas [23,24,25]. However, the histogenesis of these carcinomas remains controversial. The anaplastic carcinomas of the pancreas account for less than 10% of all pancreatic tumors; UCOGCs are a subset of these. The clinicopathological features and diagnosis of undifferentiated pancreatic carcinoma are still unclear since they are very rare. Although it is a very rare form of pancreatic carcinoma, it is more than a simple statistic for patients diagnosed with it. Therefore, it is important to pool our current understanding of the histogenesis, genetic characteristics, clinical manifestations, treatment options and prognosis of patients with undifferentiated pancreatic carcinoma. This paper aims to review the clinical characteristics and treatment experience reported in the literature.

## 2. Histogenesis

Initially, undifferentiated pancreatic carcinoma was thought to originate from acinar or mesenchymal cells. The immunohistochemistry examination in these incidences showed positive results for CD68 and vimentin [26,27]. The CD68-positive result is indicative of mononuclear histiocytic cells. A second and currently accepted origin is the epithelial route. Carcinomas of epithelial origin usually show positive CEA and keratin staining. They share some similarities with PDACs; however, they are clinically distinct in manifestation [3]. The observed histological features may vary depending on the particular histological sub-type of the undifferentiated pancreatic carcinoma. Histopathological characteristics of the substypes are summarized in Table 1.

### 2.1. Undifferentiated Carcinoma with Osteoclast-like Giant Cells

This is the most frequently reported subtype of undifferentiated pancreatic carcinoma [3,7,8,9,10,11,12,13,14,15,16,17,18,19]. UCOGCs are pathologically characterized by the presence of giant cells that resemble the giant cells of the bone; hence, they are called osteoclast-like giant cells [5]. These giant cells are diffuse, with multiple small and uniform nuclei [28]. Histological findings characteristic of UCOGCs are shown in Figure 1. Also observed are spindle-shaped cells characterized by a high mitotic index and exhibiting mesenchymal differentiation. Furthermore, histological examination shows mononuclear cells with generally typical features; however, atypical mononuclear cells may also be found [29,30].

### 2.2. Sarcomatoid Carcinoma of the Pancreas

Sarcomatoid carcinoma of the pancreas is a sub-type of undifferentiated pancreatic carcinoma that is of epithelial origin [4], also referred to as pleomorphic giant cell carcinoma of the pancreas. On histological examination, pleomorphic pancreatic carcinoma is characterized by a predominance of spindle-shaped cells, accounting for about 80% of the cells without gland formation and a growth pattern mimicking sarcoma [4,31]. There is a general absence of osteoclast-like giant cells. Pleomorphic carcinoma has both mononucleated and multinucleated giant cells. Besides spindle-shaped cells, eosinophilic pleomorphic and regular round cells are present. Histological features are shown in Figure 2.

### 2.3. Rhabdoid Pancreatic Carcinoma

Rhabdoid pancreatic carcinoma derives its name from its histological mimicking of rhabdomyosarcoma in appearance without the rhabdomyosarcoma elements. It was first used to describe a more aggressive variant of Wilms’ tumor [32]. Rhabdoid pancreatic carcinomas have distinctive cytological patterns that are of two types, as described by Agaimy et al. [33]. The pleomorphic type of rhabdoid pancreatic carcinoma, much like the sarcomatoid type, has abundant pleomorphic neoplastic giant cells with numerous eosinophilic cytoplasms. However, unlike the sarcomatoid type, they have rhabdoid inclusions. There is also an inconspicuous variation. In the second type, the monomorphic anaplastic subtype, cellular and nuclear pleomorphisms are not apparent. Instead, there are monomorphic cells that have generally large-sized vesicular nuclei with prominent nucleoli and rhabdoid inclusions [32]. Histological features are shown in Figure 3.

## 3. Molecular Characteristics

### 3.1. Immunohistochemistry

The immunohistochemistry profiles of undifferentiated pancreatic carcinoma have been reported in numerous studies. They are characterized by positive pan-cytokeratin (CKAE1/AE3) in the mononuclear pleomorphic cells [8,9,17]. Pleomorphic cells may also exhibit a low Ki-67 labeling index [8,14]. PD-L1 expression has been shown to be significantly more positive in tumor cells of undifferentiated pancreatic carcinoma than in conventional PDACs in some studies [10,34]. PD-L1 expression was positive in osteoclastic-like giant cells. In rhabdoid pancreatic carcinoma, the neoplastic cells express a loss of INI1 expression [16,34,35,36]. Osteoclastic-like giant cells are consistently non-reactive with CKAE1/AE3 (pan-cytokeratin) [8,9,12]. They have a strong reactivity to histiocytic markers, predominantly CD68. They are rarely positive for epithelial membrane antigen and A1ACT. The mononucleated cells usually lack LCA expression; however, multinucleated giant cells show membrane-accentuated positivity with LCA [6,12,31].

### 3.2. Genetic Mutations

Genetic variations in undifferentiated pancreatic carcinoma have some commonalities with PDAC. There are frequent alterations in KRAS, TP53 and CDKN2A [3,9,14,22,23]. KRAS mutations may be absent in some cases, especially in the rhabdoid subtype [30,33,36]. The presence of a KRAS mutation is a hallmark of poor prognosis. Homozygous deletions of CDKN2A and CDKN2B have been reported, and so have SMAD4 mutations [3,22]. Rhabdoid carcinoma is marked by a lack of INI1 expression [33,34,36]. INI1, also known as SMARCB1, is a core subunit of the SWI/SNF ATP-dependent chromatin remodeling complex. The deficiency of SMARCB1/INI1 is mostly associated with atypical rhabdoid/teratoid neoplasms. Loss of SMARCB1/INI1 is confined to the anaplastic monomorphic subtype of rhabdoid pancreatic carcinoma [33].

## 4. Diagnosis

Patients with undifferentiated pancreatic carcinoma usually present with generic and nonspecific signs and symptoms [9,34]. Patients may present with jaundice, itchy skin, weight loss, nausea and vomiting and fever [8,9,36]. Patients may report vague abdominal and/or back pain and gastric fullness [15,37]. In some cases, patients may be asymptomatic. Diagnosis is made based on a combination of imaging and biochemical examinations.

### 4.1. Imaging

In most cases of undifferentiated pancreatic carcinoma, computed tomography (CT), magnetic resonance imaging (MRI) and endoscopic ultrasound (EUS) are sufficient to identify the tumor, and confirmation is made by biopsy [38]. EUS fine-needle aspiration (FNA) is vital for the confirmation of the diagnosis [39]. The tumors are more frequently located in the head of the pancreas and less frequently in the tail. They may appear hypointense on dual-phase CT. On average, they are larger than PDACs at the time of diagnosis with a mean diameter of 5.3 cm [13]. In some cases, the pancreatic tumor may mimic other tumors, such as duodenal papillary carcinoma, upon endoscopic retrograde cholangiopancreatography [36]. In some cases, the neoplasm may not be apparent on CT [36]. Magnetic resonance cholangio-pancreatograpy (MRCP) can better identify any dilation of the biliary tree [40].

### 4.2. Serology

The serological findings in patients with undifferentiated pancreatic carcinoma generally feature elevated conjugated and unconjugated serum bilirubin and total bilirubin (TBIL) and liver enzymes, including aspartate aminotransferase (AST), alkaline phosphatase (ALP) and alanine aminotransferase (ALT) [9,11,35,36]. The tumor biomarker levels are nonspecific based on the current literature. Some studies have reported elevated carbohydrate antigen (CA) 19-9 [9,11,37], as well as CA 125 [35]. Other studies have reported both CA 19-9 and carcinoembryonic antigen (CEA) to be normal [16,35,36,37].

## 5. Treatment

The only curative treatment for undifferentiated pancreatic carcinoma is surgical resection. However, in many cases, patients have advanced disease at the time of diagnosis. There are no explicit/specific recommendations for the management of this particular subtype of pancreatic carcinoma. The treatment is usually informed by the experience of the clinicians treating PDAC and other types of malignant pancreatic carcinoma that are more prevalent. Patients with unresectable tumors can be treated with chemotherapy and immunotherapy.

### 5.1. Surgical Resection

Undifferentiated pancreatic carcinoma is a malignant and aggressive form of pancreatic carcinoma; therefore, radical surgical resection with negative margins (R0 resection) is the best chance for a cure. Careful and extensive evaluation by an experienced hepatobiliary surgeon or multi-disciplinary team is recommended to determine suitability for surgery. The choice of surgical protocol depends on the location of the tumor and the presence or lack of regional lymph node involvement. In the case of a tumor in the head of the pancreas without regional lymph node involvement, pylorus-preserving pancreatoduodenectomy is considered [9,11,12,13,15]. For head-of-pancreas tumors with regional lymph node involvement, extended pancreatoduodenectomy should be the choice [35]. For tumors in the body or tail of the pancreas, distal pancreatectomy with or without splenectomy is chosen, depending on the patient’s situation [14,36]. Figure 4 shows extended pancreato-duodenectomy for patient with rhabdoid pancreatic carcinoma att our center.

### 5.2. Preoperative Biliary Driange

Patients who present with jaundice and elevated conjugated, unconjugated and total bilirubin coupled with elevated liver enzymes may be at risk of liver injury. In such patients, percutaneous transhepatic bile duct drainage (PTCD) may be necessary, especially if more time is needed to assess patient eligibility for surgical resection. Preoperative biliary drainage may reduce liver failure-associated mortality [14,41]. Figure 5 shows preoperative PTCD placement to resolve jaundice in patient.

### 5.3. Chemotherapy and (Neo-)Adjuvant Therapy

Currently, there are no large-scale studies evaluating (neo-)adjuvant therapy, chemotherapy and radiotherapy in patients with undifferentiated pancreatic carcinoma. Due to the rarity of this carcinoma type, we can only gain insights from the anecdotal evidence and experience reported by various centers around the world. In patients with unresectable tumors, folfirinox has been used; however, the response seems to be suboptimal [19,36]. King et al. observed an immediate and sustained response in their patients when they switched from folfirinox to gemcitabine + nab-paclitaxel [36]. In some cases, patients initially assessed to have unresectable tumors underwent neo-adjuvant chemotherapy and later surgical resection followed by adjuvant therapy [13]. In most of the studies we looked at, patients with resectable tumors were recommended to undergo adjuvant chemotherapy. Most doctors recommended adjuvant gemcitabine-based monotherapy [11,21,42]. In other cases, a combination of gemcitabine and capecitabine was used [35,37]. Saito et al. used a combination of gemcitabine and S1 followed by oral tegafururacil for maintenance [43].

## 6. Prognosis

The prognosis of patients with undifferentiated pancreatic carcinoma is very poor, especially in patients with unresectable tumors. The median overall survival ranges between 3 and 6 months [44,45]. Even among patients who undergo surgical resection, the outcomes are not very promising either. However, longer disease-free survival and overall survival have been reported in patients who accepted postoperative adjuvant chemotherapy. Disease-free survival and overall survival of more than 5 years up to 16 years have been reported [21,43]. Although some patients who rejected postoperative adjuvant chemotherapy have shown some level of long-term survival, a better prognosis seems to be achieved in patients who accept postoperative chemotherapy. Based on the existing literature, gemcitabine monotherapy or gemcitabine combined with capecitabine may be a promising regimen for postoperative adjuvant chemotherapy [11,21,35,37,42,43].

## 7. Conclusions

Although undifferentiated pancreatic carcinomas are rare, aggressive and have a generally poor prognosis, patients should be carefully evaluated to determine if they are eligible for surgical resection. Patients with unresectable tumors may try neo-adjuvant therapy and be re-assessed. Patients with resectable tumors should be recommended postoperative adjuvant chemotherapy. This may reduce the odds of recurrence and boost the odds of long-term survival.

## Figures and Tables

**Figure 1 cancers-14-06102-f001:**
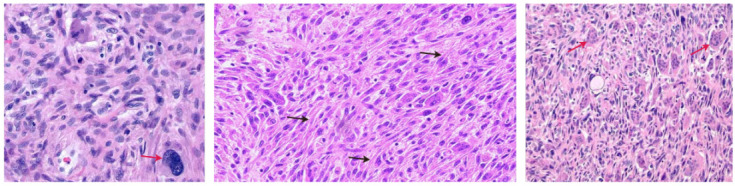
Histopathologic findings of UCOGCs showing undifferentiated cells with osteoclast-like giant cells (red arrows) and spindle-shaped cells (black arrows).

**Figure 2 cancers-14-06102-f002:**
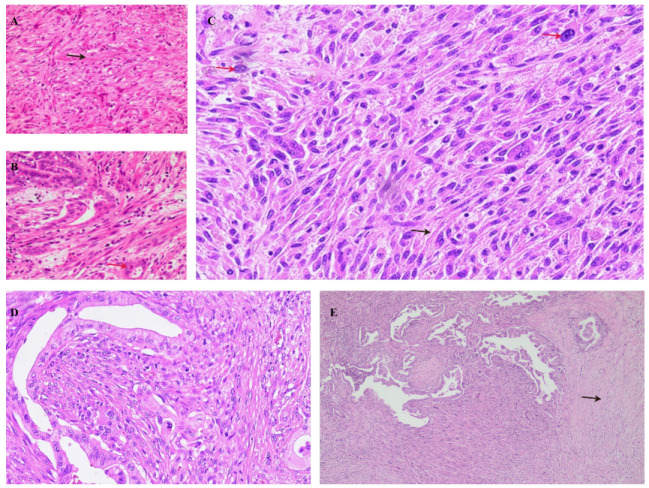
Histopathologic findings of sarcomatoid carcinoma of the pancreas showing the sarcoma-like appearance, spindle-shaped cells (black arrows) and pleomorphic cells (red arrows). (**A**,**B**) HE at 10×; (**C**,**D**) HE at 20×; (**E**) HE at 4×.

**Figure 3 cancers-14-06102-f003:**
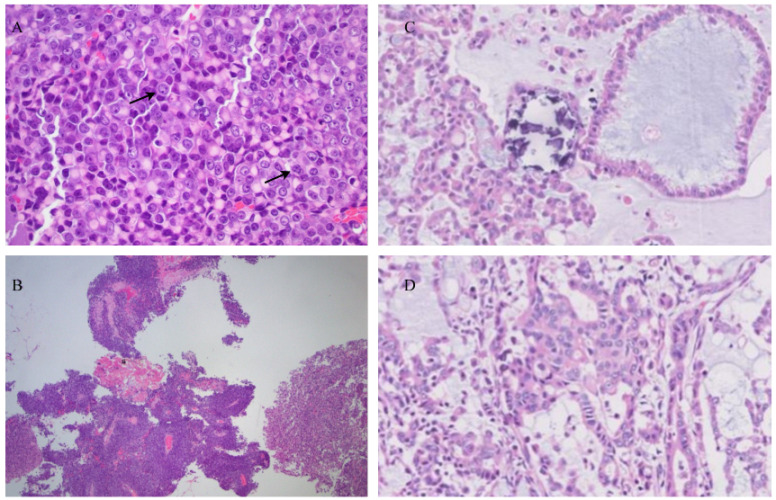
Histopathologic findings of rhabdoid pancreatic carcinoma showing the pleomorphic type of rhabdoid carcinoma with rhabdoid inclusions (black arrows). Monomorphic type of rhabdoid pancreatic carcinoma with rhabdoid inclusions. (**A**) HE stain at 400×; (**B**) HE stain at 40×; (**C**,**D**) HE stain at 10×.

**Figure 4 cancers-14-06102-f004:**
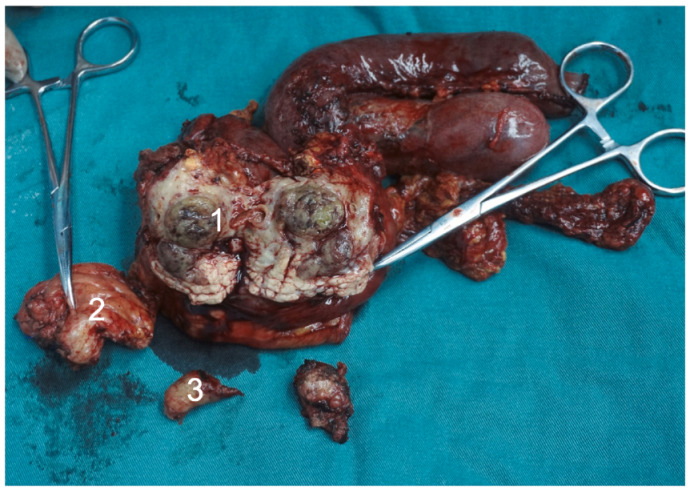
Extended pancreato-duodenectomy performed for a patient with rhabdoid pancreatic carcinoma at our center. (**1**) Tumor in the head of the pancreas. (**2**) Resected portion of the body of the pancreas to achieve negative margins. (**3**) Embolus from the main pancreatic duct.

**Figure 5 cancers-14-06102-f005:**
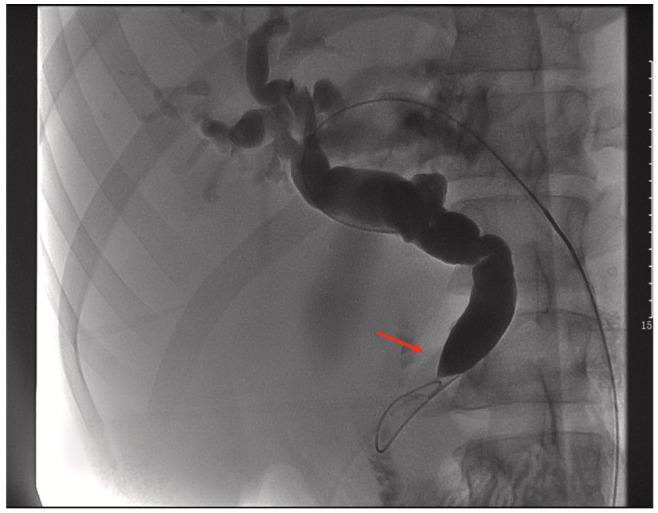
Preoperative PTCD. Imaging showing that the common bile duct (CBD) leading into the ampulla of Vater was compressed and obstructed, similar to a beak (red arrow).

**Table 1 cancers-14-06102-t001:** Histopathological characteristics of undifferentiated pancreatic tumors.

UCOGC	Sarcomatoid	Rhabdoid
Spindle-shaped cells	Spindle-shaped cells	Mimicking rhabdamyoscarcoma
Giant osteoclast-like cells	Pleomorphic cells	Monomorphic cells
Mononuclear cells	Mononuclear and multinuclear cells	Prominent nuclei
High mitotic index	Glandular formation	Cellular and nuclear pleomorphism may be absent
	Mimics sarcoma	Rhabdoid inclusions
	Osteoclast-like giant cells are usually absent	Loss of SMARCAB1/INI1

## Data Availability

Not applicable.

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
