# Peer review of "Undifferentiated Pancreatic Carcinomas, Clinical Features and Therapeutic Options: What We Know"

_cancers, 2022, doi:10.3390/cancers14246102_

Round 1

Reviewer 1 Report

This review is focused on a very low incidence subtype of pancreatic tumor, Undifferentiated Carcinoma of the Pancreas. In recent years much attention is being paid to identify the different subtypes of pancreatic cancer since the responses to therapies are totally different depending on the specific molecular characteristics of each subtype. It should be noted that the fact that there are several different subtypes in a cancer with such a low incidence makes it very difficult to establish generalities for each type of PDAC. In any case, if the scientific community wants to improve the survival rates of this terrible disease, approaches such as this one are necessary to understand the particularities of each PDAC subtype. Therefore, the topic discussed seems to me to be of great interest to the community focused on the study of pancreatic cancer, although this may mean that the audience is not too wide.

·         For the histological images, it would be appreciated if you could indicate with arrows the most characteristic details, as well as identify the magnification at which the images are made.

·         It would be very beneficial for the reader to have a summary table identifying the main characteristics of each of the histopathological subtypes of the undifferentiated pancreatic carcinomas.

·         What is the incidence amongst the 3 types?

·         Is there no correlation between histological and molecular features? Just as the three subtypes have been discussed at the histological level (1- Undifferentiated carcinoma with osteoclast-like giant cells; 2- Sarcomatoid carcinoma of the pancreas and 3- Rhabdoid pancreatic carcinoma), it would be very interesting to know if there is a correlation with some particular molecular characteristics. Moreover, all this information could be collected in the same table to which I referred above.

·         Is there any difference in the overall survival amongst the 3 types? And among Undifferentiated pancreatic carcinomas and pancreatic ductal adenocarcinomas?

·         For the majority of PDAC patients (80% or more), surgery is not an option as the disease is far too advanced and thus inoperable. The serology is quite vague and very similar to pancreatic ductal adenocarcinomas. What are the real options to identify these subtypes?

·         The treatments listed are mainly the same applied for pancreatic ductal adenocarcinomas.

·         If the diagnosis, prognosis, and treatment are so similar to what is done today for pancreatic adenocarcinoma, there is no way to identify these types of patients unless a biopsy is performed, which is not always advisable.

Author Response

Thank you for your valued feedback. We have endeavored to address all the comments you raised and made changes where needed. We these sufficiently address the issues raised.

Please find a point by point response to your comments below:

For the histological images, it would be appreciated if you could indicate with arrows the most characteristic details, as well as identify the magnification at which the images are made.

We have added arrows to show key features where needed. The magnifications of the images have also been added to the descriptions.

It would be very beneficial for the reader to have a summary table identifying the main characteristics of each of the histopathological subtypes of the undifferentiated pancreatic carcinomas.

Table 1 has beed added to the manuscript to summarize the key characteristics of the subtypes of undifferentiated pancreatic carcinoma.

What is the incidence amongst the 3 types?

The undifferentiated pancreatic carcinomas are very rare. They are under anaplastic pancreatic carcinomas which have an incidence rate of < 10% (included in the introduction section). The individual incidence rates of the particular subtypes range between < 1% to about 5%.

Is there no correlation between histological and molecular features? Just as the three subtypes have been discussed at the histological level (1- Undifferentiated carcinoma with osteoclast-like giant cells; 2- Sarcomatoid carcinoma of the pancreas and 3- Rhabdoid pancreatic carcinoma), it would be very interesting to know if there is a correlation with some particular molecular characteristics. Moreover, all this information could be collected in the same table to which I referred above. 

We have indicated in the table 1 the characteristics of the subtypes. There overlap between the subtypes which some difference. For example loss of SMARCB1/INI1 is characteristic of the Rhabdoid subtype while the sacromatoid subtype mimics sarcoma and osteoclast-like giant cells are often absent. 

Is there any difference in the overall survival amongst the 3 types? And among Undifferentiated pancreatic carcinomas and pancreatic ductal adenocarcinomas?

Undifferentiated pancreatic carcinoma are very aggressive and have a generally poor prognosis. They are often unresectable and most patients die within 6 months (Included in the introduction section).  However it is difficult to establish a direct statistical comparison due the rarity of the undifferentiated pancreatic carcinomas. The general understanding is that they have a poorer prognosis than PDAC.

For the majority of PDAC patients (80% or more), surgery is not an option as the disease is far too advanced and thus inoperable. The serology is quite vague and very similar to pancreatic ductal adenocarcinomas. What are the real options to identify these subtypes?

Like PDAC, most undifferentiated pancreatic carcinomas are advanced at the time of diagnosis. US-FNA is helpful in making an accurate diagnosis. However, it may still be difficult to fully determine the size of the tumor prior to operation as was the case with one of our patients.

The treatments listed are mainly the same applied for pancreatic ductal adenocarcinomas.

There are no standard guidelines for undifferentiated pancreatic carcinomas because they are very rare. Most centers make an informed decision based on experience treating other pancreatic carcinomas. For this reason, we discussed treatment options we and other centers have used to provide some insight.

If the diagnosis, prognosis, and treatment are so similar to what is done today for pancreatic adenocarcinoma, there is no way to identify these types of patients unless a biopsy is performed, which is not always advisable.

It is true that the diagnosis and treatment options are quite similar. WHO classifies these undifferentiated pancreatic carcinomas under adenocarcinomas. However, in clinical practice, they are much more difficult to diagnose than PDAC. They may be misdiagnosed as acinar cell carcinoma and the true size of the tumor is often difficult to determine prior to operation. While we have a lot experience and literature to draw from for PDAC, the literature on undifferentiated pancreatic carcinomas is not much. The object of this review is to provide some insights in the rare cases where they re encountered.

Although these are rare tumors, to our patients, they are not just statistic but a major life challenge for which both the clinician and patient seek some literature to better understand the outlook of the disease.

Reviewer 2 Report

My congratulations to the authors for their interesting article. The authors should include in all topics personal contributions based on their experience.

Author Response

Thank you for you valued feedback. We have added the author contributions to the manuscript as you suggested.

Round 2

Reviewer 1 Report

Authors have addressed all my suggestions. 

Reviewer 2 Report

My congratulations to the authors for improving their manuscript.